# Constructing Invariant and Equivariant Operations by Symmetric Tensor Network

## Abstract

Design of neural networks that incorporate symmetry is crucial for geometric deep learning. Central to this effort is the development of invariant and equivariant operations. This work presents a systematic method for constructing valid invariant and equivariant operations. It can handle inputs and outputs in the form of Cartesian tensors with different ranks, as well as spherical tensors with different types. In addition, our method features a graphical representation utilizing the symmetric tensor network, which simplifies both the proofs and constructions related to invariant and equivariant functions. We also show how to apply this method to design the equivariant interaction message for the geometry graph neural network and general neural network models incorporating symmetry.

## 1 Introduction

Many scientific problems involve systems with 3D geometric structures, such as molecules and materials. For these systems, we can always describe them using 3D coordinates. However, physical quantities in the natural world do not depend on any specific coordinate system. They exhibit invariance or equivariance under changes of coordinates, such as rotations and translations. Therefore, when performing machine learning tasks involving physical quantities, it is advantageous in data efficiency and generalization to incorporate invariance or equivariance into the hypothesis neural networks (Geiger & Smidt, 2022).

Designing the invariant or equivariant operations in the neural networks is a crucial step in developing the invariance/equivariance machine learning models. For equivariant geometry graph neural networks (EGNN) with vector features, the equivariant operations on vector features are typically vector summation $v_1 + v_2$ (Satorras et al., 2021; Schütt et al., 2021; Deng et al., 2021) and vector product $v_1 \times v_2$ (Le et al., 2022). For Tensor Field Network (TFN) style with higher-type (order) spherical tensors features, the equivariant operations most used are the tensor product (TP) operations (Thomas et al., 2018; Weiler et al., 2018). In addition, recent works also use the higher-rank Cartesian tensors as the equivariant feature in the message passing, in which the typical equivariant operations used are tensor contraction and summation of tensors (Wang et al., 2024).

In this work, we developed a systematic tool capable of constructing $SO(3)$ invariance and equivariance operations given the specified forms of input and output, which include tuples of various rank Cartesian tensors and various types spherical tensors. The main tool we used is the symmetric tensor network (Singh et al., 2010; 2011; Singh & Vidal, 2012), which is widely used in quantum many-body systems. Combining the classical invariance theory (Weyl, 1946), we developed a framework for building generators of invariant functions, which we call *tensor network generators*, designed for a specific input format. We can obtain the equivariant operations corresponding to specified input and output quantity forms by calculating the derivatives of the tensor network generators. Based on the method we developed, we demonstrate how it can be used to construct invariant and equivariant operators in geometric graph neural networks and how to construct general neural network models incorporating symmetry.

## 2 PRELIMINARIES

### 2.1 TENSOR NETWORK

The tensor networks have been proven to be a powerful graphical language and computational tool across multiple disciplines. The roots of this diagrammatic notation can be traced back to the work of Roger Penrose in the 1970s Penrose et al. (1971). These structured decompositions of high-dimensional tensors into networks of lower-dimensional tensors were originally developed in the context of quantum many-body physics (Vidal, 2003; Verstraete & Cirac, 2006; Schollwöck, 2011), but have since found widespread applications in machine learning (Levine et al., 2018; Hayashi et al., 2019; Ma & Solomonik, 2022; Wang et al., 2025), quantum computing (Pan et al., 2022; Pan & Zhang, 2022), applied mathematics (Oseledets, 2011), and beyond. We further discuss related literature to situate our work within the broader field in Section 6.

We first introduce the formalism of tensors, which are the building blocks of tensor networks. A tensor $T$ is a multi-dimensional array. We can denote its elements as $T_{i_1,i_2,...,i_n} \in \mathbb{R}^{I_1 \times I_2 \times \cdots \times I_n}$, where the $n$ is the rank of the tensor and $I_k$ is the dimension size of index $i_k$. The tensor also has a graphical representation. As shown in Fig.1(a), a tensor can be represented by a node with legs, where each leg corresponds to an index of the tensor. A vector can be represented by a one-leg node, and a matrix can be represented by a two-leg node.

A tensor network is a collection of tensors defined above. The legs connected between nodes are the indices needed to be summed over, which is called contraction. Therefore, the tensor network can be contracted to a single tensor, the index of which corresponds to the open leg of the tensor network. As shown in Fig.1(b), the tensor network describes the contraction of tensor $A$ and $B$, i.e, matrix multiplication. Furthermore, we can also give the graphical representation of the derivative of the tensor network. As shown in Fig.1(c), the derivative of a tensor network with respect to a specific tensor $T$ (which appears only once in the network) is a tensor network where tensor $T$ is removed.

Actually, a tensor network represents a certain decomposition of a high-rank tensor. As shown in Fig.1(d), a rank-N tensor is decomposed to rank-2 and rank-3 tensors, where this decomposition is called tensor train decomposition (Oseledets, 2011) (or matrix product state (Vidal, 2003; Verstraete & Cirac, 2006) in the quantum many-body physics community).

### 2.2 GROUP INVARIANCE AND EQUIVARIANCE

We can give the definition of group invariance and equivariance functions as follows:

**Definition 2.1.** *Let $G$ be a group which acts on linear spaces $V_1, \ldots, V_n$ over field $F$ by certain linear representation. An invariant function $f : \bigoplus_i V_i \to F$ is a multi-variable function such that for each $g \in G$,*

$$f(g \cdot \boldsymbol{x}_1, \cdots, g \cdot \boldsymbol{x}_n) = f(\boldsymbol{x}_1, \cdots, \boldsymbol{x}_n) \tag{1}$$

**Definition 2.2.** *Let $G$ be a group which acts on linear spaces $V_1, \ldots, V_n, U_1, \ldots, U_m$ over $F$ by certain linear representation. An equivariant function $f : \oplus_j V_j \to \oplus_i U_i$ in a multi-variable function such that for each $g \in G$,*

$$f^i(g \cdot \boldsymbol{x}_1, \cdots, g \cdot x_n) = g \cdot f^i(\boldsymbol{x}_1, \cdots, \boldsymbol{x}_n) \tag{2}$$

where the $\cdot$ means the group action on a linear space. For example, the inputs of the equivariance functions can be 3D coordinates of each atom in a molecule, and the outputs can be the force of each atom. In the remaining part of the paper, we mainly focus on $SO(3)$ group and are restricted to the real case $F = \mathbb{R}$.

### 2.3 SYMMETRIC TENSOR NETWORK

A symmetric tensor (Singh et al., 2010; 2011; Singh & Vidal, 2012) is a tensor that is invariant under a group action in the space of each of its indices.

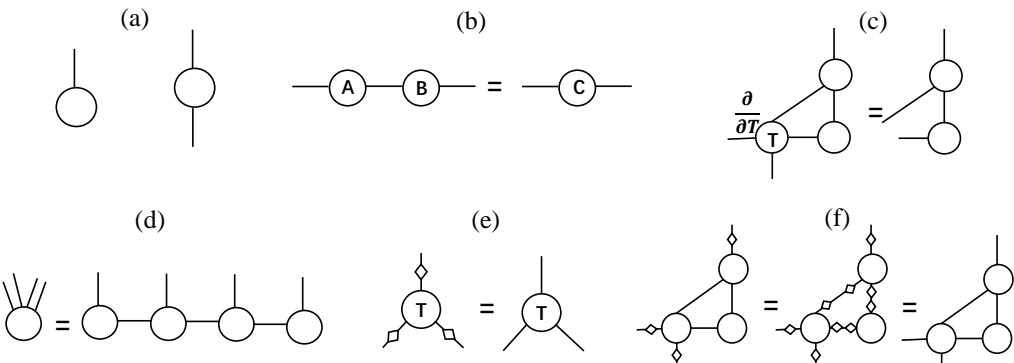

Figure 1: Tensor networks and symmetric tensor networks. (a) Graphical representation of a rank-1 tensor (vector) and a rank-2 tensor (matrix). (b) The contraction of tensor $A$ and $B$, this is the matrix multiplication $C_{ik} = \sum_j A_{ij} B_{jk}$. (c) The derivative of a tensor network with respect to a specific tensor $T$, the result of which is a tensor network where tensor $T$ is removed. (d) Tensor train decomposition, namely, a rank-N tensor is decomposed to rank-2 and rank-3 tensors. (e) The graphical illustration of the equation $\forall g \in G : \prod_i \rho_i(g)_{\alpha_i, \beta_i} T_{\alpha_1, \ldots, \alpha_n} = T_{\beta_1, \ldots, \beta_n}$. (f) Tensor networks that consist of symmetric tensors are also symmetric tensors as a whole. We first insert identity $\rho_i(g)\rho_i(g)^T = I$ on the contracted leg. Since every tensor in the network is symmetric, the tensor networks as a whole are also symmetric.

**Definition 2.3.** *Let $T_{i_1, \ldots, i_n} \in \mathbb{R}^{I_1 \times \cdots \times I_n}$ be a tensor, and $\rho_i : G \mapsto GL(I_i, \mathbb{R})$ be the group representation on space of $i$-th index. $(T_{i_1, \ldots, i_n}, (\rho_1, \ldots))$ is called a (group)* **symmetric tensor** *iff*

$$\forall g \in G : \prod_i \rho_i(g)_{\alpha_i, \beta_i} T_{\alpha_1, \ldots, \alpha_n} = T_{\beta_1, \ldots, \beta_n} \tag{3}$$

This can be illustrated by Fig. 1(e). A symmetric tensor network is a collection of symmetric tensors defined above. Fig. 1(f) provides a graphical representation that tensor networks that consist of symmetric tensors are also symmetric tensors as a whole. Thus, the contraction operation preserves tensor symmetry (Singh et al., 2010). This fundamental property allows us to employ a more restrictive formation for tensors with predefined symmetries, namely, constructing the tensor network exclusively from symmetric tensors (Singh et al., 2010). We give more details about the symmetric tensor network in the Appendix A.

### 2.4 CARTESIAN AND SPHERICAL TENSORS

A *Cartesian tensor* of rank $r$ is an element of the tensor product space $T \in (\mathbb{R}^3)^{\otimes r}$. Given a rotation $R \in SO(3)$, the group acts on $T$ by rotating each index, $(R \cdot T)_{i_1 \ldots i_r} = R_{i_1 j_1} \cdots R_{i_r j_r} T_{j_1 \ldots j_r}$.

A *spherical tensor* of type $l \in \{0, 1, 2, \ldots\}$ is an element of the irreducible $SO(3)$ representation space $V_l$ of dimension $2l + 1$. The spherical tensor $T$ with type $l$ has components $T_m$ and under a rotation $R$ these components transform according to $(R \cdot T)_m = \sum_{m'=-l}^l D_{mm'}^{(l)}(R) T_{m'}$, where $D^{(l)}(R)$ is the Wigner $D$-matrix of degree $l$. Spherical tensors are the basic building blocks of irreducible representations of $SO(3)$.

## 3 TENSOR NETWORK GENERATORS

The polynomials of $n$ variables input $\boldsymbol{x} = (\boldsymbol{x}_1, \ldots, \boldsymbol{x}_n)$, denoted by $F[V]$, actually form a mathematical structure called algebra. It's easy to see that the invariant polynomials form a subalgebra denoted by $F[V]^G$. By Hilbert's finiteness theorem (Hilbert, 1890; 1893), $F[V]^G$ is finitely generated when $G$ is a finite group or compact Lie group (including the case that $G = SO(3)$). In other words, any invariant polynomial $f(\boldsymbol{x})$ can be written as a polynomial $q$ of a set of polynomials $\{g_1, \ldots, g_n\}$. These polynomials, called the generators of $F[V]^G$, are independent of the $f$. More generally, the Weierstrass approximation theorem states that any continuous invariant function can be approximated by an invariant polynomial (Weierstrass, 1885). It follows, therefore, that any invariant function can be approximated by a function of these generators $\{g_1, \ldots, g_n\}$. We'll give a systematic method to construct the generators of $\mathbb{R}[V]^{SO(3)}$ by tensor network, which we call **tensor network generators**.

### 3.1 VECTOR INPUTS

Firstly, let's consider the simplest case, where the inputs are 3D vectors $\boldsymbol{x}_1, \ldots, \boldsymbol{x}_n \in \mathbb{R}^3$. In this case, the input space is $V = \mathbb{R}^{3n}$. Weyl(Weyl, 1946) proved that

**Lemma 3.1.** *The set of generators of $\mathbb{R}[V]^{SO(3)}$ is $\{\boldsymbol{x}_i \cdot \boldsymbol{x}_j, (\boldsymbol{x}_i \times \boldsymbol{x}_j) \cdot \boldsymbol{x}_k\}$. Therefore any invariant polynomial takes the form of $f(\boldsymbol{x}) = q(\{\boldsymbol{x}_i \cdot \boldsymbol{x}_j, (\boldsymbol{x}_i \times \boldsymbol{x}_j) \cdot \boldsymbol{x}_k\})$.*

This lemma not only provides a finite set of generators of 3D vector inputs, but can also be used to greatly simplify the structure of a symmetric tensor, which is useful in treating the other input form, which we discuss later.

**Lemma 3.2.** *Each $SO(3)$ symmetric tensor $T \in \mathbb{R}^{3 \times 3 \times \cdots \times 3}$ is generated by identity tensor $\delta_{ij}$ and Levi-Civita tensor $\epsilon_{ijk}$. That is to say, each $SO(3)$ symmetric tensor $T$ is a linear combination of tensors, each of which is the tensor product of $\delta_{ij}$ and $\epsilon_{ijk}$.*

*Furthermore, if the rank of $T$ is even, then $T$ is a linear combination of tensors, each of which is the tensor product of $\delta_{ij}$. Otherwise, $T$ is a linear combination of tensors, each of which is the tensor product of $\delta_{ij}$ together with exactly one $\epsilon_{ijk}$.*

We give the proof in Appendix B. The symmetric tensor $\delta_{ij}$ and $\epsilon_{ijk}$ can be represented by

$$\delta_{ij} : \frown \qquad \epsilon_{ijk} : \overset{|}{\frown} \tag{4}$$

It should be noted that these $SO(3)$ symmetric tensors whose indices take the 3D representation, along with their characteristic properties, are also known as isotropic tensors (Jeffreys, 1973) in the classical invariance theory.

### 3.2 CARTESIAN TENSOR INPUTS

Next, let's consider a more general case, where the input $\boldsymbol{x}_1, \ldots, \boldsymbol{x}_n \in \mathbb{R}^{3 \times 3 \times \cdots \times 3}$ are Cartesian tensors with various ranks whose indices take the 3D representation of $SO(3)$. For this case, we can construct the tensor network generators in the following way,

**Theorem 3.3.** *Let $\boldsymbol{x}_1, \ldots, \boldsymbol{x}_n$ be input Cartesian tensors whose indices take the 3D representation of SO(3). Let $V$ be the input space. Then $\mathbb{R}[V]^{SO(3)}$ is generated by the contraction of connected tensor network formed by $\boldsymbol{x}_1, \ldots, \boldsymbol{x}_n$ (multiplicity is allowed) together with identity tensor $\delta_{ij}$ and at most one Levi-Civita tensor $\epsilon_{ijk}$.*

The proof of Theorem 3.3 is given in Appendix C. When all inputs $\boldsymbol{x}_1, \ldots, \boldsymbol{x}_n$ are vectors, the tensor network generators are $\{\boldsymbol{x}_i \cdot \boldsymbol{x}_j, (\boldsymbol{x}_i \times \boldsymbol{x}_j) \cdot \boldsymbol{x}_k\}$, which is the same as in Lemma 3.1. In this case, it's easy to see that $\{\boldsymbol{x}_i \cdot \boldsymbol{x}_j, (\boldsymbol{x}_i \times \boldsymbol{x}_j) \cdot \boldsymbol{x}_k\}$ is a minimal set of generators. However, if the inputs $\boldsymbol{x}_1, \ldots, \boldsymbol{x}_n$ contain tensors of higher rank, the tensor-network generators are not minimal. Since by Hilbert's finiteness theorem (Hilbert, 1890; 1893), we have a finite set of generators $(g_1, \ldots, g_m)$ which are generated by a finite subset of tensor-network generators. That means only a finite subset of tensor-network generators suffices to generate the whole $\mathbb{R}[V]^{SO(3)}$. However, it's very difficult to determine the exact subset. In practice, we can take the subset of tensor network polynomials with degree $\leq D$, which gives a finite-degree approximation to the minimal set of generators. We also noted that recent work (Gregory et al., 2024) constructs the equivariant polynomial function with Cartesian tensor inputs in a similar way, which uses the group averaging property of the orthogonal groups (Jeffreys, 1973).

### 3.3 SPHERICAL TENSOR INPUTS

Irreducible representations of $SO(3)$ group can be labeled by a non-negative integer $l$. The representation $l$ is $(2l + 1)$-dimensional, and the common 3D representation is just the representation $1$. In this section, we consider inputs $\boldsymbol{x}_1, \ldots, \boldsymbol{x}_n$ of irreducible representation spaces, that is, each $\boldsymbol{x}_i \in V_i$ where $V_i$ is a real linear space of an irreducible $SO(3)$ representation $l_i$.

A rank-$r$ tensor whose indices take the 3D representation of $SO(3)$ is a reducible representation and can be reduced as follows.

$$(1)^{\otimes l} = (l) \oplus (l-1)^{d_{l,l-1}} \oplus \cdots \oplus (s)^{d_{l,s}} \oplus \cdots \oplus (0)^{d_{l,0}} \tag{5}$$

where $(1)^{\otimes l}$ denotes the $l$-fold tensor-product representation space i.e. $(1)^{\otimes l} = \underbrace{(1) \otimes \cdots \otimes (1)}_{l}$ and $(s)^d = \underbrace{(s) \oplus \cdots \oplus (s)}_{d}$. The value of $d_{l,s}$ is given in Appendix D.

For each representation $l$, we can define an $SO(3)$-symmetric tensor $P_l$ which projects from the space $(1)^{\otimes l}$ to the space $l$. The exact form we constructed of $P_l$ is given in Appendix E. For $P_l$, we have

$$\tag{6}$$

Where the numbers on the edges represent the irreducible representation type of the vector space associated with that index. Let $\boldsymbol{x}$ be a variable of representation $l$, we define $P_l(\boldsymbol{x})$ to be

. Then $P_l(x)$ becomes a tensor in the space $(1)^{\otimes l}$, which can be used to construct tensor network generators.

**Theorem 3.4.** *Let $\boldsymbol{x}_1, \ldots, \boldsymbol{x}_n$ be input variables that take the irreducible representation $l_1, \ldots, l_n$ of SO(3). Let $V$ be the input space. Then $\mathbb{R}[V]^{SO(3)}$ is generated by the contraction of connected tensor network formed by $P_{l_1}(\boldsymbol{x}_1), \ldots, P_{l_n}(\boldsymbol{x}_n)$ (multiplicity is allowed) together with at most one Levi-Civita tensor $\epsilon_{ijk}$.*

The proof is given in Appendix F.

### 3.4 GENERAL TENSOR INPUTS

In the most general case, we would expect that the inputs $\boldsymbol{x}_1, \ldots, \boldsymbol{x}_n$ are vectors in $SO(3)$ of general representation spaces. For example, $\boldsymbol{x}_i$ may be of representation $(1) \oplus (3) \oplus (7)$. The construction of tensor network generators is similar to the last section of spherical tensor inputs, except for each $\boldsymbol{x}_i$ (which is of representation $\omega_i$), we should choose projector $P_{\omega_i}$ which projects from the space $(1)^{\otimes r_i}$ to the space $\omega_i$, where $\omega_i$ is a sub-representation of $(1)^{\otimes r_i}$. See Appendix E for the exact form of $P_\omega$.

### 3.5 GENERALIZATION TO EQUIVARIANT FUNCTIONS

Actually, we can always construct an equivariant function $f$ from an invariant function (Blum-Smith & Villar, 2023). We have

**Lemma 3.5.** *Given an invariant function $f : \bigoplus_j V_j \oplus \bigoplus_i U_i \to F$ with input $\boldsymbol{x}_1, \ldots, \boldsymbol{x}_n, \boldsymbol{y}_1, \ldots, \boldsymbol{y}_m$ in space $V_1, \ldots, V_n, U_1, \ldots, U_m$, we can always construct an equivariant function $T_{\mathrm{up}}(f) : \bigoplus_j V_j \to \bigoplus_i \bar{U}_i$, where $G$ acts on $\bar{U}_i$ by the dual representation of $U_i$, by defining*

$$T_{\mathrm{up}}(f)^i(\boldsymbol{x}_1, \cdots, \boldsymbol{x}_n) = \left. \frac{\partial f(\boldsymbol{x}_1, \cdots, \boldsymbol{x}_n, \boldsymbol{y}_1, \ldots, \boldsymbol{y}_m)}{\partial \boldsymbol{y}_i} \right|_{\boldsymbol{y}_1 = \cdots = \boldsymbol{y}_m = 0} \tag{7}$$

*where we have chosen a natural set of basis for $U_i$ and the corresponding dual basis for $\bar{U}_i$. Besides, any equivariant function can be obtained in this way.*

We give the proof of the above Lemma and more details about obtaining equivariant functions from the invariant functions in Appendix G. From the Lemma 3.5 and tensor network calculations, we can obtain the following theorem (We give the proof of the theorem in Appendix G.),

**Theorem 3.6.** *Let $\boldsymbol{x}_1, \ldots, \boldsymbol{x}_n$ be input variables in space $V_1, \ldots, V_n$ and $\boldsymbol{y}_1, \ldots, \boldsymbol{y}_m$ be output variables in space $U_1, \ldots, U_m$, each equivariant function $h : \bigoplus_j V_j \to \bigoplus_i \bar{U}_i$ can be expressed as*

$$h^i(\boldsymbol{x}_1, \ldots, \boldsymbol{x}_n) = \sum_j q_j^i(g_1, \ldots, g_n) t_j^i \tag{8}$$

*where $q_j^i$ are functions, $g_1, \ldots, g_n \in \mathbb{R}[V]^{SO(3)}$ are tensor network generators of $\boldsymbol{x}_1, \ldots, \boldsymbol{x}_n$, and $t_j^i$ is the tensor networks (labeled by $j$) which are obtained by removing the output variables $\boldsymbol{y}_i$ from the tensor network generators of $\boldsymbol{x}_1, \ldots, \boldsymbol{x}_n, \boldsymbol{y}_i$, in which the $\boldsymbol{y}_i$ variable appears exactly once.*

## 4 CONSTRUCTING INVARIANT AND EQUIVARIANT OPERATIONS FOR GEOMETRY GRAPH NEURAL NETWORK

In this section, we will show how to use the framework developed above to construct the equivariant operations of the geometry GNN. Geometry GNNs are built on graph-structured data with the 3D geometric information. For the node feature $\mathbf{h}_i$ of node $i$ and $\mathbf{h}_j$ of its neighbour $j \in N(i)$ in layer $l$, the interaction message $\mathbf{m}_{ij}^{l+1}$ for them is $\mathbf{m}_{ij}^{l+1} = f_m\left(\mathbf{h}_i^l, \mathbf{h}_j^l\right)$, where the map $f_m$ is a learnable function. Then the interaction message $\mathbf{m}_{ij}$ for all the neighbour $j \in N(i)$ are aggregated by a permutation invariant function $\bigoplus_{j \in \mathcal{N}(i)}$, such as sum and mean, which is used to update the node feature $\mathbf{h}_i$ of node $i$ according to another learnable function $\mathbf{h}_i^{l+1} = f_u\left(\mathbf{h}_i^l, \bigoplus_{j \in \mathcal{N}(i)} \mathbf{m}_{ij}^l\right)$. For the equivariant geometry GNN, the functions $f_m$ and $f_u$ need to be $SO(3)$ equivariant functions. Many of the equivariant geometry GNN researches focus on designing novel equivariant functions $f_m$ and $f_u$. The feature $\mathbf{h}_i$ and message $\mathbf{m}_{ij}$ can be cartesian tensor (Wang et al., 2024) or spherical tensor (Thomas et al., 2018; Weiler et al., 2018).

## 4.1 Constructing invariant and equivariant operations for Cartesian tensor feature

For Cartesian tensor feature $\mathbf{h}$ and message $\mathbf{m}_{ij}$, we consider the tensor rank to be 0 (scalar), 1 (vector), 2 (matrix), or higher. Here, we show how to construct the functions $f_m$ with Cartesian tensor feature and message.

We represent the scalar invariants as fully contracted networks involving the input features $\mathbf{h}_i, \mathbf{h}_j$ and the message $\mathbf{m}_{ij}$. In the diagrams below, nodes represent the input/output quantities, and connecting lines represent the contraction of indices with the identity tensor $\delta_{ij}$ or the Levi-Civita tensor $\epsilon_{ijk}$. Specifically, the red lines denote the output space corresponding to the indices of the message $\mathbf{m}_{ij}$ before the final contraction to a scalar. By taking the derivative with respect to $\mathbf{m}_{ij}$ (removing the red lines), we obtain the equivariant functions. For simplicity, we just construct the tensor networks that $\mathbf{h}$ and $\mathbf{m}_{ij}$ appear at most once. More complicated operations can be constructed in the similar way.

**Vector input and vector output:** We set $\mathbf{h}_i = \mathbf{u}, \mathbf{h}_j = \mathbf{v}, \mathbf{m}_{ij} = \mathbf{w}$ and $\mathbf{u}, \mathbf{v}, \mathbf{w} \in \mathbb{R}^3$. The invariances we can construct are

(9)

Eq.(9) just is $u_i(v_i)w_j\delta_{ij}$ [1] and $\varepsilon_{ijk}u_iv_jw_k$ We can obtain the equivariants just by removing the output tensor such that

(10)

which is precisely $\mathbf{u}(\mathbf{v})$ and $\mathbf{u} \times \mathbf{v}$. These operations are typically the equivariant operations used in the EGNN-style Satorras et al. (2021); Schütt et al. (2021); Deng et al. (2021); Le et al. (2022).

**Matrix input and matrix output:** For matrix features, we can set $\mathbf{h}_i = \mathbf{A}, \mathbf{h}_j = \mathbf{B}, \mathbf{m}_{ij} = \mathbf{C}$ and $\mathbf{A}, \mathbf{B}, \mathbf{C} \in \mathbb{R}^{3 \times 3}$. The invariances we can construct are

(11)

which are $\mathrm{Tr}(\mathbf{C}^{(T)})$, $\mathrm{Tr}(\mathbf{A}^{(T)}\mathbf{C}^{(T)})/\mathrm{Tr}(\mathbf{B}^{(T)}\mathbf{C}^{(T)})$, and $\mathrm{Tr}(\mathbf{A}^{(T)}\mathbf{B}^{(T)}\mathbf{C}^{(T)})/\mathrm{Tr}(\mathbf{B}^{(T)}\mathbf{A}^{(T)}\mathbf{C}^{(T)})$ [2]. We can get equivariances by derivation,

(12)

which is $I$, $\mathbf{A}^{(T)}/\mathbf{B}^{(T)}$, $\mathbf{A}^{(T)}\mathbf{B}^{(T)}/\mathbf{B}^{(T)}\mathbf{A}^{(T)}$. These equivariant operations correspond to tensor contraction and summation in the context of higher-rank Cartesian tensor features (Wang et al., 2024).

**Matrix input and vector output:** If we set the matrix features $\mathbf{h}_i = \mathbf{A}, \mathbf{h}_j = \mathbf{B}$ and $\mathbf{A}, \mathbf{B} \in \mathbb{R}^{3 \times 3}$ and the interaction message $\mathbf{m}_{ij} = \mathbf{v} \in \mathbb{R}^3$. The invariances we can construct are as follows,

(13)

---

[1] $u_i(v_i)w_j\delta_{ij}$ means $u_iw_j\delta_{ij}$ and $v_iw_j\delta_{ij}$.

[2] Here, $\mathrm{Tr}(\mathbf{C}^{(T)})$ means $\mathrm{Tr}(\mathbf{C})$ and $\mathrm{Tr}(\mathbf{C}^T)$. $\mathrm{Tr}(\mathbf{A}^{(T)}\mathbf{C}^{(T)})$ means $\mathrm{Tr}(\mathbf{AC})$, $\mathrm{Tr}(\mathbf{A}^T\mathbf{C}^T)$, $\mathrm{Tr}(\mathbf{A}^T\mathbf{C})$, $\mathrm{Tr}(\mathbf{AC}^T)$. The meaning of $\mathrm{Tr}(\mathbf{A}^{(T)}\mathbf{B}^{(T)}\mathbf{C}^{(T)})$ and $\mathrm{Tr}(\mathbf{B}^{(T)}\mathbf{A}^{(T)}\mathbf{C}^{(T)})$ are also similar.

We can get the corresponding equivariances by derivation,

$$\text{(14)}$$

where the first diagram $A_{ij}\epsilon_{ijk}$ is the axial vector, which consists of the elements of the antisymmetric part of $A_{ij}$.

### 4.2 CONSTRUCTING INVARIANT AND EQUIVARIANT OPERATIONS FOR SPHERICAL TENSOR FEATURE

We can also use this framework to express the equivariant function for spherical tensor inputs and outputs. We set the equivariant function with inputs feature $\mathbf{h}_i = \boldsymbol{a}$ of representation $l_a$, $\mathbf{h}_j = \boldsymbol{b}$ of representation $l_b$ and output message $\mathbf{m}_{ij} = \boldsymbol{c}$ of representation $l_c$. As in Appendix H, we consider the tensor network generator

$$g(\boldsymbol{a}, \boldsymbol{b}, \boldsymbol{c}) = \qquad \text{(15)}$$

where $l_{ab} = \frac{l_a + l_b - l_c}{2}, \quad l_{bc} = \frac{l_b + l_c - l_a}{2}, \quad l_{ca} = \frac{l_c + l_a - l_b}{2}$.

In our framework, we can obtain the equivariant function as follows.

$$\boldsymbol{c} = T_{\text{up}}\left(Ag(\boldsymbol{a}, \boldsymbol{b}, \boldsymbol{c})\right) = A \qquad \text{(16)}$$

Here, $A$ is an arbitrary scale factor. This is precisely the TP operations of the TNF-style proposed in works (Thomas et al., 2018; Weiler et al., 2018) (we give more detail in Appendix H).

## 5 CONSTRUCTING EQUIVARIANT MACHINE LEARNING MODEL

We can also use the framework we developed to construct $SO(3)$ invariant and equivariant neural network model. For $SO(3)$ invariant neural network model, we can construct the functions as

$$f_{\text{inv}}(\boldsymbol{x}) = q(g_1(\boldsymbol{x}), \ldots, g_m(\boldsymbol{x})), \qquad \text{(17)}$$

where $g_1, \ldots, g_m$ are tensor network generators and $q$ is a general neural network model, as shown in Fig. 2(a). In addition, according to the Lemma 3.5 and Theorem 3.6, we can always construct an equivariant neural network model from an invariant neural network model using the transformation $T_{\text{up}}$, as shown in Fig. 2(b). In this way, we can decouple the symmetry constraints from the neural network architectures.

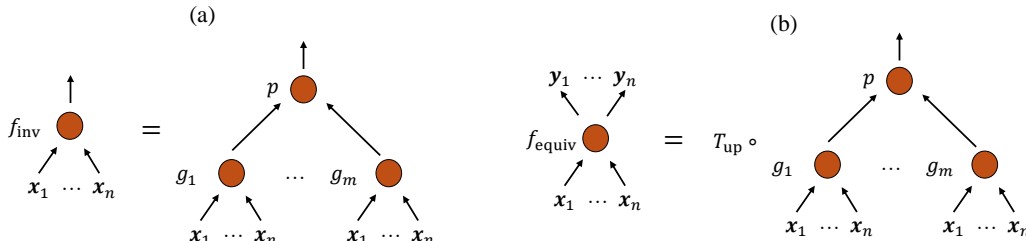

Figure 2: (a) The general form of $SO(3)$ invariant neural network model $f_{\text{inv}}$ by composing a general neural network $p$ with $g_1, \ldots, g_m$, which are tensor network generators. (b) The general form of $SO(3)$ equivariant neural network model by composing a general neural network $p$ with $g_1, \ldots, g_m$, which are tensor network generators, and then applying $T_{\text{up}}$.

## 6 RELATED WORK

**Invariant and equivariant functions**: Works (Villar et al., 2021; Gregory et al., 2024) show how to construct the $O(N)$ invariant and equivariant polynomials for Cartesian tensors in theory based on Weyl's classical invariance theory (Weyl, 1946). Work Pearce-Crump (2023) constructed linear and $O(N)/SO(N)$ equivariant functions of Cartesian tensors based on Brauer algebra Brauer (1937). Furthermore, works (Blum-Smith & Villar, 2023) show how to get equivariant functions from the derivative of invariant polynomials based on the method of B. Malgrange.

**Geometry graph neural networks.** Existing works for geometric GNNs can be categorized by their strategy for constructing equivariant operations. Scalar-based approaches generate features primarily through invariant operations by inner products and the equivariant operations typically by vector summations and product (Satorras et al., 2021; Schütt et al., 2021; Deng et al., 2021). Tensor Product-based approaches, pioneered by TFN (Thomas et al., 2018) and 3D Steerable CNNs (Weiler et al., 2018), use the higher-type spherical tensors as feature and construct equivariant operations using Clebsch-Gordan tensor products. This line of work also includes Le et al. (2022); Brandstetter et al. (2022); Liao & Smidt (2023). Work Batatia et al. (2022) have advanced these methods from modeling two-body interactions to capturing complex many-body interactions. Recently, higher-rank Cartesian tensors are also used as the equivariant feature in the message passing (Wang et al., 2024). Theoretical works (Dym & Maron, 2021; Joshi et al., 2023) have further explored the expressivity and universality of geometric GNNs.

**Tensor network:** Early contributions (Singh et al., 2010; 2011; Singh & Vidal, 2012) described how to incorporate the $SU(2)$ symmetry for tensor networks states for quantum many-body systems. More recently, Works (Li et al., 2024) use the fusion diagram (a graphical representation of the successive Clebsh-Gordan products) to construct the $SO(3)$ equivariant blocks. Works (Hodapp & Shapeev, 2024) used the symmetric tensor network for constructing the machine-learning interatomic potentials. In addition, work (Kunisky et al., 2024) shows how to construct the $O(N)$ invariant and equivariant functions for Cartesian tensors by tensor network.

In this work, we establish a general framework for constructing concrete tensor network generators that are applicable to any given input and output, which consist of tuples of Cartesian tensors of various ranks and spherical tensors of various types.

## 7 CONCLUSION AND DISCUSSION

This work introduces a general framework for constructing invariant and equivariant operations, designed to handle various data forms, including tuples of Cartesian tensors of different ranks and spherical tensors of different types. This framework is particularly useful for building symmetric operations within geometric graph neural networks. Determining the optimal subset of these operators for specific graph tasks is a promising direction for future work, particularly in combination with Neural Architecture Search Elsken et al. (2019). More broadly, it can be applied to any machine learning model that requires symmetry constraints by first generating fundamental equivariant and invariant quantities, which can then be leveraged by conventional neural networks for learning.

Although our formal results are stated for $SO(3)$, the framework extends naturally to $O(3)$ by augmenting the tensor network representation with a parity label. We can assign to every input, output, and structural tensor a parity label (odd or even) , so that each tensor network has a global parity given by the product of all its constituent parities. $O(3)$-invariant polynomials are then obtained by restricting the corresponding tensor networks with total parity even. We can follow Theorem 3.6 and remove the output tensor from such an invariant network yields an equivariant polynomial that correctly respects the parity of the target output quantity.

In future work, we can utilize low-rank, structured tensor networks to parameterize complex equivariant operations, thereby improving sample efficiency and reducing computational complexity.

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

## A  SYMMETRIC TENSOR NETWORKS

The contraction of a tensor network yields a single resultant tensor. When the contracted tensor is constrained by a specific global symmetry, we can assume that every constituent tensor in the original network also satisfies the same symmetry condition, which is termed as symmetric tensors (Singh et al., 2010; 2011; Singh & Vidal, 2012). This deduction is justified by the invariant property that the contraction operation preserves the symmetry characteristics of symmetric tensors.

The vector space $V$ associated with each index of a symmetric tensor constitutes a representation space of the group, which admits the following decomposition:

$$V \cong \bigoplus_s d_s V_s \cong \bigoplus_s (D_s \otimes V_s), \tag{18}$$

where $V_s$ is the irreducible representation space for the $s$-th representation of the group, $d_s$ denotes the multiplicity of this representation, and $D_s$ stands for the $d_s$-dimensional degeneracy space.

The basis for the $r$-th index of the tensor can be constructed using the decomposition described in Eq. (18). In the decomposition of the representation space $V = \bigoplus_s (D_s \otimes V_s)$ for some index, the basis vectors are parameterized by the triplet $(s, \alpha_s, m_s)$, where $s$ labels the irreducible representations of the group, $\alpha_s = 0, 1, \cdots d_s - 1$ indexes the degeneracy space $D_s$ of dimension $d_s$, and $m_s$ corresponds to the internal basis of the irreducible representation space $V_s$.

Under this basis, a tensor satisfies the symmetry condition if it meets the following criteria at each rank:

- **Zeroth-rank (Scalar):** A scalar is trivially a symmetric tensor.
- **First-rank:** Non-zero elements must reside in the trivial irreducible representation ($s = 0$).
- **Second-rank:** Non-zero elements require both indices to belong to the conjugate irreducible representation space. The tensor must take the form:

$$T_{(s_0, \alpha_{s_0}, m_{s_0}), (s_1, \alpha_{s_1}, m_{s_1})} = P_{(s_0, \alpha_{s_0}), (s_1, \alpha_{s_1})} \delta_{s_0, s_1} \begin{pmatrix} & s_0 & \\ m_{s_0} & & m_{s_1} \end{pmatrix}, \tag{19}$$

  where the brackets denote the Wigner 1-jm symbol (Wigner, 1993).
- **Third-rank:** The tensor must satisfy the following structure:

$$T_{(s_0, \alpha_{s_0}, m_{s_0}), (s_1, \alpha_{s_1}, m_{s_1}), (s_2, \alpha_{s_2}, m_{s_2})} = P_{(s_0, \alpha_{s_0}), (s_1, \alpha_{s_1}), (s_2, \alpha_{s_2})} \begin{pmatrix} s_0 & s_1 & s_2 \\ m_0 & m_1 & m_2 \end{pmatrix}, \tag{20}$$

  with the brackets representing the Wigner 3-jm symbol (Wigner, 1993).
- **Higher-rank:** The tensor must be decomposable into contractions of multiple third-rank or lower-rank symmetric tensors.

## B  PROOF OF LEMMA 3.2

*Proof of Lemma 3.2.* Let $T$ be an $SO(3)$-symmetric tensor whose indices take the 3D representation of $SO(3)$, and $\boldsymbol{x}_i$ be variables each in $\mathbb{R}^3$. We can define a $SO(3)$-symmetric polynomial

$$f(\boldsymbol{x}) = \sum_{i_1, \ldots, i_n} T_{i_1, \ldots, i_n} (\boldsymbol{x}_1)_{i_1} \cdots (\boldsymbol{x}_n)_{i_n} \tag{21}$$

By Lemma 3.1, we have $f(\boldsymbol{x}) = \sum_i c_i p_i$ where each $p_i$ is product of elements in $\{\boldsymbol{x}_i \cdot \boldsymbol{x}_j, (\boldsymbol{x}_i \times \boldsymbol{x}_j) \cdot \boldsymbol{x}_k\}$ and $c_i$ is the coefficients. Taking derivative of $\boldsymbol{x}_1, \ldots, \boldsymbol{x}_n$ on both sides, we can see that $T$ is of the form of finite sum $T = \sum_i c_i T_i$, where $c_i \in \mathbb{R}$ and each $T_i$ is the tensor product of $\delta_{ij}$ and $\epsilon_{ijk}$.

Considering the parity of the rank, there is odd (even) $\epsilon_{ijk}$ in each $T_i$ if the rank of $T$ is odd (even). Notice that

$$\epsilon_{ijk}\epsilon_{lmn} = \delta_{il}(\delta_{jm}\delta_{kn} - \delta_{jn}\delta_{km}) - \delta_{im}(\delta_{jl}\delta_{kn} - \delta_{jn}\delta_{kl}) + \delta_{in}(\delta_{jl}\delta_{km} - \delta_{jm}\delta_{kl}) \tag{22}$$

Then tensor product of odd number of $\epsilon_{ijk}$ reduces to one $\epsilon_{ijk}$. The tensor product of even number of $\epsilon_{ijk}$ reduces to product and sum of the tensor $\delta_{ij}$. $\square$

## C    PROOF OF THEOREM 3.3

*Proof of Theorem 3.3.* It's easy to see that each invariant polynomial is a finite sum of homogeneous invariant polynomials. Therefore, we only need to study homogeneous invariant polynomials. Let $p$ be a homogeneous invariant polynomial. Then we can write $p(\boldsymbol{x}_1, \ldots, \boldsymbol{x}_n)$ as a tensor network contraction.

$$p(\boldsymbol{x}_1, \ldots, \boldsymbol{x}_n) = \tag{23}$$

where multiplicity of $\boldsymbol{x}_1, \ldots, \boldsymbol{x}_n$ is allowed. Since $p$ is invariant, for $g \in SO(3)$ we have

$$p(\boldsymbol{x}_1, \ldots, \boldsymbol{x}_n) = p(U(g)^{\otimes r_1}\boldsymbol{x}_1, \ldots, U(g)^{\otimes r_n}\boldsymbol{x}_n) \tag{24}$$

$U$ is the 3D representations of $SO(3)$ and $r_i$ is the rank of $\boldsymbol{x}_i$. That is,

$$= \tag{25}$$

Taking derivative of $\boldsymbol{x}_1 \cdots \boldsymbol{x}_1 \cdots \boldsymbol{x}_n \cdots \boldsymbol{x}_n$ on both sides, we have

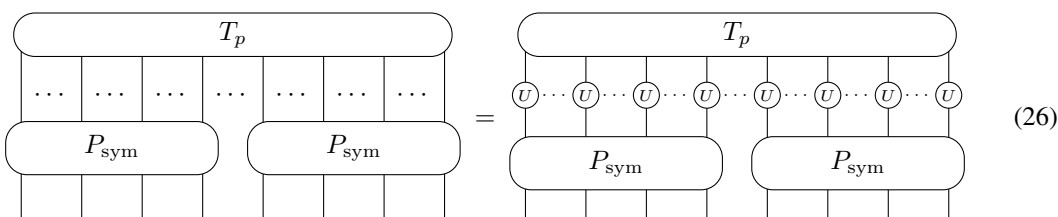

$$= \tag{26}$$

where $P_{\text{sym}}$ is the projection to symmetric subspace under permutation within identical $\boldsymbol{x}_i$s.

$$P_{\text{sym}}{}^{p_{1,1}\ldots p_{t_i,r_i}}_{q_{1,1}\ldots q_{t_i,r_i}} = \frac{1}{t_i!} \sum_{\sigma \in S_{t_i}} \prod_j \prod_k \delta^{p_{j,k}}_{q_{\sigma(j),k}} \tag{27}$$

where $t_i$ is the multiplicity of $x_i$, and $\sigma$ take value in all permutation of $t_i$ elements.

It's easy to see that $P_{\text{sym}}$ commutes with $U^{\otimes r_i t_i}$

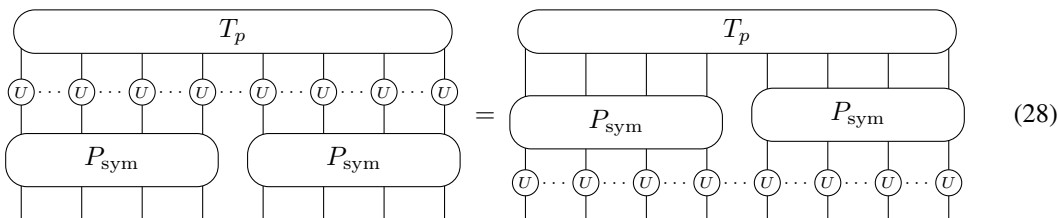

$$= \tag{28}$$

Therefore we may define

$$
\tag{29}
$$

Combining Eq.(26) and Eq.(28), it's easy to see that $T_p'$ is an $SO(3)$-symmetric tensor. By Lemma 3.2, $T_p'$ is a linear combination of the tensor product of delta tensors and at most one Levi-Civita tensor. For polynomial $p(\boldsymbol{x}_1, \ldots, \boldsymbol{x}_n)$, we can always use the partially permutation symmetrization of the tensor $T_p$ as its coefficients, that is,

$$
p(\boldsymbol{x}_1, \ldots, \boldsymbol{x}_n) = \tag{30}
$$

Therefore, $p(\boldsymbol{x}_1, \ldots, \boldsymbol{x}_n)$ is a linear combination of the contraction of tensor networks formed by $\boldsymbol{x}_1, \ldots, \boldsymbol{x}_n$(multiplicity is allowed) together with at most one Levi-Civita tensor $\epsilon_{ijk}$. Since contraction of disconnected tensor network is product of the contraction of each component, $\mathbb{R}[V]^{SO(3)}$ is generated by the contraction of connected tensor network formed by $\boldsymbol{x}_1, \ldots, \boldsymbol{x}_n$ (multiplicity is allowed) together with at most one Levi-Civita tensor $\epsilon_{ijk}$. $\qquad\square$

## D  DECOMPOSITION OF TENSOR PRODUCT REPRESENTATION $(1)^{\otimes l}$

We have

$$
(1)^{\otimes l} = (l)^{d_{l,l}} \oplus (l-1)^{d_{l,l-1}} \oplus \cdots \oplus (s)^{d_{l,s}} \oplus \cdots \oplus (0)^{d_{l,0}} \tag{31}
$$

By counting the dimension of $L_z$ eigen-spaces [3], we have

$$
d_{l,s} = \sum_{i=s}^{\lfloor \frac{s+l}{2} \rfloor} \frac{l!}{i!(s+l-2i)!(i-s)!} - \sum_{i=s+1}^{\lfloor \frac{s+l+1}{2} \rfloor} \frac{l!}{i!(s+l+1-2i)!(i-s-1)!}. \tag{32}
$$

Particularly, $d_{l,l} = 1$, $d_{l,l-1} = l - 1$ and $d_{l,l-2} = \frac{l(l-1)}{2}$. When $l$ is small, the exact decomposition is listed below

$$
(1)^{\otimes 2} = (2) \oplus (1) \oplus (0) \tag{33}
$$

$$
(1)^{\otimes 3} = (3) \oplus (2)^2 \oplus (1)^3 \oplus (0) \tag{34}
$$

$$
(1)^{\otimes 4} = (4) \oplus (3)^3 \oplus (2)^6 \oplus (1)^6 \oplus (0)^3 \tag{35}
$$

$$
(1)^{\otimes 5} = (5) \oplus (4)^4 \oplus (3)^{10} \oplus (2)^{15} \oplus (1)^{15} \oplus (0)^6 \tag{36}
$$

---

[3] $L_x, L_y, L_z$ is the orbital angular momentum operator in x, y, z direction, respectively(Zee, 2016).

# E   THE EXACT FORM OF PROJECTOR $P_l$ AND $P_\omega$

For $l$ representation, we have basis vector $|m\rangle$ ($-l \le m \le l$). The $l$ sub-representation in $(1)^{\otimes l}$ satisfies

$$\sqrt{\frac{(2l)!(l-m)!}{(l+m)!}}|m\rangle = (L_-)^{l-m}|l\rangle = \left(\sum_i L_{-,i}\right)^{l-m}|1,\ldots,1\rangle \tag{37}$$

where $L_\pm = L_x \pm iL_y$ is the ladder operator.[4]

Then

$$|m\rangle = \sqrt{\frac{(l+m)!}{(2l)!(l-m)!}}(\sqrt{2})^{l-m}\sum_{(s_i),\sum s_i=m}\frac{(l-m)!}{2^{d(s)}}|s_1,\ldots,s_l\rangle \tag{38}$$

$$= \sqrt{\frac{(l-m)!(l+m)!2^{l-m}}{(2l)!}}\sum_{(s_i),\sum s_i=m}\frac{1}{2^{d(s)}}|s_1,\ldots,s_l\rangle \tag{39}$$

where $s_i = 0, \pm 1$ and $d(s)$ is the number of $-1$ in $(s) = (s_1, \ldots, s_l)$.

Therefore

$$(P_l)_m^{(s)} = \frac{1}{2^{d(s)}}\sqrt{\frac{(l-m)!(l+m)!2^{l-m}}{(2l)!}}\delta_{\sum s_i,m} \tag{40}$$

To get $P_\omega$, we need to construct the orthogonal matrix $O_l$ of the transformation $(1)^{\otimes l} \to (l_1) \oplus (l_2) \oplus \cdots$. If we define $Q_l$ as the orthogonal matrix of the transformation $(l) \otimes (1) \to (l+1) \oplus (l) \oplus (l-1)$. We can infer $O_l$ by induction:

$$O_1 = Q_1 \tag{41}$$

$$\tag{42}$$

$Q_l$ can be obtained directly from the CG coefficients of $\mathfrak{so}(3)$ Lie algebra.

# F   PROOF OF THEOREM 3.4

*Proof of Theorem 3.4.* Let $p$ be a homogeneous invariant polynomial. Similar to the proof of Theorem 3.3, we have

$$p(\boldsymbol{x}_1, \ldots, \boldsymbol{x}_n) = \tag{43}$$

---

[4]$L_-|m\rangle = \sqrt{(l-m)(l+m+1)}|m-1\rangle$ and $L_-|-l\rangle = 0$, where $|m\rangle$ is the standard basis of $l$ representation. Especially, for 1 representation, $L_-|1\rangle = \sqrt{2}|0\rangle$, $L_-|0\rangle = \sqrt{2}|-1\rangle$ and $L_-|-1\rangle = 0$ (Zee, 2016).

where $T_p'$ is an $SO(3)$-symmetric tensor. Inserting identities, we have

$$p(\boldsymbol{x}_1, \ldots, \boldsymbol{x}_n) = \tag{44}$$

where the tensor network

$$\tag{45}$$

is $SO(3)$-symmetric and the free legs are of 3D representation. By Lemma 3.2, the contraction of this tensor network is linear combination of tensor product of delta tensors and at most one Levi-Civita tensor. Therefore $p(\boldsymbol{x}_1, \ldots, \boldsymbol{x}_n)$ is linear combination of contraction of tensor network formed by $P_{l_1}(\boldsymbol{x}_1), \ldots, P_{l_n}(\boldsymbol{x}_n)$ (multiplicity is allowed) together with at most one Levi-Civita tensor $\epsilon_{ijk}$. Since contraction of disconnected tensor network is product of the contraction of each component, $\mathbb{R}[V]^{SO(3)}$ is generated by the contraction of connected tensor network formed by $P_{l_1}(\boldsymbol{x}_1), \ldots, P_{l_n}(\boldsymbol{x}_n)$ (multiplicity is allowed) together with at most one Levi-Civita tensor $\epsilon_{ijk}$. $\square$

## G OBTAINING THE EQUIVARIANT FUNCTIONS FROM THE INVARANT FUNCTIONS

We can give the proof of Lemma 3.5 as following:

*Proof of Lemma 3.5.* Since $f$ is invariant, for each $g \in G$,

$$T_{\mathrm{up}}(f)^i(\boldsymbol{x}_1, \cdots, \boldsymbol{x}_n)_\alpha = \left.\frac{\partial f(\boldsymbol{x}_1, \cdots, \boldsymbol{x}_n, \boldsymbol{y}_1, \cdots, \boldsymbol{y}_m)}{\partial(\boldsymbol{y}_i)_\alpha}\right|_{\boldsymbol{y}_1 = \cdots = \boldsymbol{y}_m = 0} \tag{46}$$

$$= \left.\frac{\partial f(g \cdot \boldsymbol{x}_1, \cdots, g \cdot \boldsymbol{x}_n, g \cdot \boldsymbol{y}_1, \cdots, g \cdot \boldsymbol{y}_m)}{\partial(\boldsymbol{y}_i)_\alpha}\right|_{\boldsymbol{y}_1 = \cdots = \boldsymbol{y}_m = 0} \tag{47}$$

$$= \sum_\beta \left.\frac{\partial f(g \cdot \boldsymbol{x}_1, \cdots, g \cdot \boldsymbol{x}_n, \boldsymbol{y}_1', \cdots, \boldsymbol{y}_m')}{\partial(\boldsymbol{y}_i')_\beta}\right|_{\boldsymbol{y}_1' = \cdots = \boldsymbol{y}_m' = 0} \frac{\partial(\boldsymbol{y}_i')_\beta}{\partial(\boldsymbol{y}_i)_\alpha} \tag{48}$$

$$= \sum_\beta T_{\mathrm{up}}(f)^i(g \cdot \boldsymbol{x}_1, \cdots, g \cdot \boldsymbol{x}_n)_\beta \rho_i(g)_{\beta\alpha} \tag{49}$$

where $\boldsymbol{y}_i' = g \cdot \boldsymbol{y}_i$.

Therefore

$$T_{\text{up}}(f)^i(g \cdot \boldsymbol{x}_1, \cdots, g \cdot \boldsymbol{x}_n)_\alpha = \sum_\beta T_{\text{up}}(f)^i(\boldsymbol{x}_1, \cdots, \boldsymbol{x}_n)_\beta \rho_i(g^{-1})_{\beta\alpha} \tag{50}$$

$$= \sum_\beta \bar{\rho}_i(g)_{\alpha\beta} T_{\text{up}}(f)^i(\boldsymbol{x}_1, \cdots, \boldsymbol{x}_n)_\beta \tag{51}$$

$$= (g \cdot T_{\text{up}}(f)^i(\boldsymbol{x}_1, \cdots, \boldsymbol{x}_n))_\alpha \tag{52}$$

$T_{\text{up}}(f)$ is equivariant.

To prove that any equivariant function can be obtained in this way, we consider the following construction. Given an equivariant function $f : \bigoplus_j V_j \to \bigoplus_i U_i$ with input $\boldsymbol{x}_1, \ldots, \boldsymbol{x}_n$ in space $V_1, \ldots, V_n$ and output $\boldsymbol{y}_1, \ldots, \boldsymbol{y}_m$ in space $U_1, \ldots, U_m$, we can construct an invariant function $T_{\text{down}}(f) : \bigoplus_j V_j \oplus \bigoplus_i \bar{U}_i \to F$, where $G$ acts on $\bar{U}_i$ by the dual representation of $U_i$, by defining

$$T_{\text{down}}(f)(\boldsymbol{x}_1, \cdots, \boldsymbol{x}_n, \boldsymbol{y}_1, \ldots, \boldsymbol{y}_m) = \sum_i \left\langle f^i(\boldsymbol{x}_1, \cdots, \boldsymbol{x}_n), \boldsymbol{y}_i \right\rangle \tag{53}$$

where we have choose a natural set of basis for $U_i$ and the corresponding dual basis for $\bar{U}_i$, $\langle \cdot, \cdot \rangle$ denotes the natural function $U_i \times \bar{U}_i \to \mathbb{R}$.

By simple deduction, one can show that $T_{\text{up}} \circ T_{\text{down}}(f) = f$. Therefore, any equivariant function $f$ can be constructed by $T_{\text{up}}(h)$ where $h = T_{\text{down}}(f)$. $\qquad\square$

Notice that for $SU(2)$ and $SO(3)$, each representation is similar to its dual. The transformation $T_{\text{up}}$ and $T_{\text{down}}$ can be pictorially expressed by Fig.3.

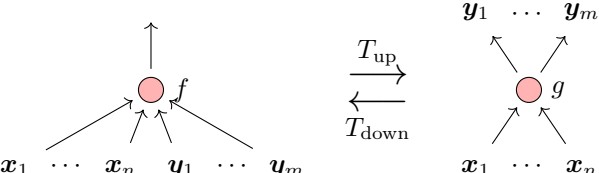

Figure 3: $T_{\text{up}}$ transforms an invariant function into an equivariant function. $T_{\text{down}}$ transforms an equivariant function into an invariant function.

Therefore, we can prove Theorem 3.6 as following,

*Proof of Theorem 3.6.* We classify the tensor network generators into three types

1. $G_1$: tensor network generators that only contain $\boldsymbol{x}_i$

2. $G_2$: tensor network generators that contain exactly one $\boldsymbol{y}_i$

3. $G_3$: tensor network generators that contain more than one $\boldsymbol{y}_i$

According to Lemma 3.5

$$h^i(\boldsymbol{x}_1, \cdots, \boldsymbol{x}_n) = \left. \frac{\partial f(\{\boldsymbol{g}_u\}, \{\bar{\boldsymbol{t}}_k^v\}, \{\boldsymbol{s}_w\})}{\partial \boldsymbol{y}_i} \right|_{\bar{\boldsymbol{t}}_j^v = \boldsymbol{s}_w = 0} \tag{54}$$

where $\boldsymbol{g}_u \in G_1$, $\bar{\boldsymbol{t}}_k^v \in G_2$ ($v$ means the generator contains exactly one $\boldsymbol{y}_v$), $\boldsymbol{s}_w \in G_3$.

It's easy to see that

$$h^i(\boldsymbol{x}_1, \cdots, \boldsymbol{x}_n) = \sum_j \left. \frac{\partial f(\{\boldsymbol{g}_u\}, \{\bar{\boldsymbol{t}}_k^v\}, \{\boldsymbol{s}_w\})}{\partial \bar{\boldsymbol{t}}_j^i} \right|_{\bar{\boldsymbol{t}}_j^v = \boldsymbol{s}_w = 0} \frac{\partial \bar{\boldsymbol{t}}_j^i}{\partial \boldsymbol{y}_i} \tag{55}$$

$$= \sum_j q_j^i(\{\boldsymbol{g}_u\}) \boldsymbol{t}_j^i \tag{56}$$

where $q_j^i(\{\boldsymbol{g}_u\})$ is a function that represents $\left. \frac{\partial f(\{\boldsymbol{g}_u\}, \{\bar{\boldsymbol{t}}_k^v\}, \{\boldsymbol{s}_w\})}{\partial \bar{\boldsymbol{t}}_j^i} \right|_{\bar{\boldsymbol{t}}_j^v = \boldsymbol{s}_w = 0}$, and $\boldsymbol{t}_j^i$ is $\bar{\boldsymbol{t}}_j^i$ with $\boldsymbol{y}_i$ missing. $\quad \square$

## H  REPRESENT SPHERICAL TENSOR EQUIVARIANT FUNCTION BY TENSOR NETWORK

The TP operations can be expressed by

$$\boldsymbol{c}_t = \sum_{rs} M_{rst} \boldsymbol{a}_r \boldsymbol{b}_s \tag{57}$$

where the type of $\boldsymbol{a}$, $\boldsymbol{b}$ and $\boldsymbol{c}$ are $l_a$, $l_b$ and $l_c$ respectively, $1 \leq r \leq 2l_a + 1, 1 \leq s \leq 2l_b + 1, 1 \leq t \leq 2l_c + 1$ and $M$ is a rank-3 symmetric tensor, whose indices are of representation $l_a, l_b$ and $l_c$.

By theory of symmetric tensor in appendix A, $M_{rst}$ proportional $C_{rst}$, which is the unique symmetric projection tensor $(l_a) \otimes (l_b) \to (l_c)$ (also called CG coefficients). In other words, we have

$$\boldsymbol{c}_t = \sum_{rs} A C_{rst} \boldsymbol{a}_r \boldsymbol{b}_s \tag{58}$$

Following the construction method of Theorem 3.4, we have

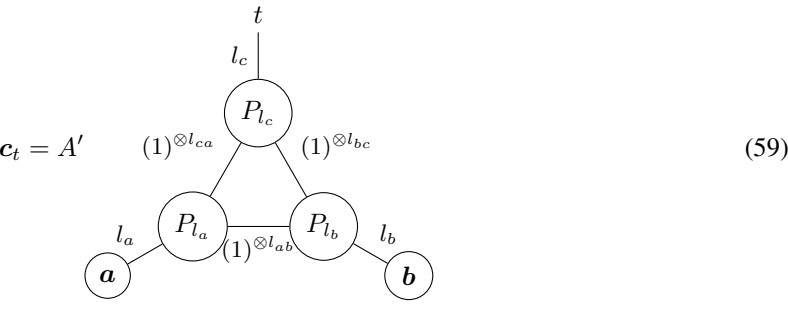

$$\boldsymbol{c}_t = A' \tag{59}$$

We define

$$C'_{rst} = \quad (1)^{\otimes l_{ca}} \quad \begin{array}{c} t \\ (l_c) \mid \\ \boxed{P_{l_c}} \\ (l_a) \boxed{P_{l_a}} \underset{(1)^{\otimes l_{ab}}}{\quad} \boxed{P_{l_b}} \, (l_b) \\ r \qquad\qquad s \end{array} \quad (1)^{\otimes l_{bc}} \quad . \tag{60}$$

To prove that our construct is equivalent to the TP operation, we only need to prove that $C'_{rst}$ is non-zero (obviously $C'_{rst} = 1$ when $r, s, t$ are of the highest weight) and is proportional to $C_{rst}$ by a factor independent of $a, b, c$, which is clear since $C'_{rst}$ and $C_{rst}$ are both symmetric rank-3 tensor with indices of representation $(l_a), (l_b), (l_c)$, and the symmetric rank-3 tensor with indices of representation $(l_a), (l_b), (l_c)$ is unique up to rescaling.

## I   THE USE OF LARGE LANGUAGE MODELS

We use large language models to polish and refine some sentences in this article.