# OpenReview forum: "Constructing Invariant and  Equivariant Operations by Symmetric Tensor Network"
_ICLR.cc/2026/Conference — Submitted to ICLR 2026_

### Official Review · Reviewer_5UcL · 2025-10-17

**Soundness:** 3
**Presentation:** 2
**Contribution:** 2
**Rating:** 4
**Confidence:** 4

**Summary:**

This paper claims to introduce a graphical tensor network framework for constructing SO(3)-invariant and equivariant operations for Cartesian tensors and spherical tensors. In particular, they claim to build generators of invariant functions which they apply in the geometric graph neural network domain, specifically in designing the equivariant interaction message between nodes.

**Strengths:**

- I commend the authors on contributing a practical framework for SO(3)-equivariance that is based upon results that are found in classical invariant theory.
- The formulation in terms of tensor networks is useful in terms of diagrammatically representing the operations.
- The ability to model both Cartesian tensors and spherical tensors using tensor networks is potentially significant and should result in a wide range of applications.
- Their experiment in Section 5 clearly demonstrates the potential of their approach.

**Weaknesses:**

- I think the theoretical contribution is very limited. All of the theory used (the results in Section 3) appear in the classical invariant theory literature. In my view only the reformulation of these results in tensor framework notation is original, but I think that originality (in terms of theoretical contribution) is very limited.
- A crucial point for me is the following, which I'd like the authors to address: I think the related work section is 1) in the wrong place - it should be included wholly in the main part of the paper and 2) is missing what I think is key literature that needs to be compared against. For example, in my view, the paper [1] surpasses the theoretical results in this paper: it came up with a complete, explicit basis for modelling all O(n) and SO(n) invariant and equivariant maps between tensors based on the paper [2], which is stronger than the results given in section 3 of the current paper. Indeed, firstly, the current paper only looks at SO(3), whereas [1] looks at all of SO(n). Secondly, the results used in section 3 only tell you what kinds of invariants generate the algebra, and moreover create redundant generating functions, so there is no minimal set of generators necessarily. In contrast, [1] gives a fully constructive description and tells you specifically how to build the invariants exactly. I definitely think a comparison between the results here and those in [1] need to be included explicitly in the paper.
- I think section 4.1 needs to be much clearer, as the link between the tensor network notation introduced in section 2 is not made clear in its application in this section. I also think the notation is somewhat ambiguous and sloppy (e.g in 9), despite the subscript.
- Given my view that the strength of this paper is solely in its practical application, I think having only one application of their method is not sufficient. I would have liked to have seen more applications where the authors' framework could be applied, especially since their framework works for both Cartesian tensors and spherical ones.
- I appreciate that the authors have used LLMs to improve the grammar and refine sentences in the paper. However, I think there are still far too many typos and grammatical errors in the paper which makes it more difficult to read. I recommend that the authors go through this paper again carefully, correcting as many of them as they can in order to improve the paper's readability.

[1] Pearce-Crump, E. - Brauer's Group Equivariant Neural Networks (ICML 2023)

[2] Brauer, R. - On Algebras Which Are Connected with the Semisimple Continuous Groups (1937)

**Questions:**

- I would like the authors to comment on the weaknesses I have listed, especially regarding the paper [1] above.
- I would like them to explain/demonstrate what sorts of problems their practical framework can be applied to, since for me this is their main contribution (which is only demonstrated so far by the one experiment in the paper itself).

---

> ### Author Response · Authors · 2025-11-24
> **Response to Reviewer 5UcL Part 1/2**
>
> We thank the reviewer for the careful reading and the detailed comments. Below we address each of the raised weaknesses and questions.
>
> >**W1.** Theoretical Contribution.
>
> We thank the reviewer for this comment. We respectfully clarify that our work is **not** a reformulation of existing  the classical invariant theory results. The classical invariant theory (e.g., Weyl) identifies generators for vector inputs. Our core contribution is establishing a systematic framework capable of constructing any $SO(3)$ invariant and equivariant map for a wide class of input–output types, including: (1)Tuples of arbitrary Cartesian tensors(Sec. 3.2), our method handles inputs of any rank (e.g., matrices, rank-3 tensors), going beyond the vector-only cases often discussed in the invariant theory. (2) Tuples of arbitrary spherical irreps (Sec. 3.3), we provide explicit tensor network generators for inputs labeled by arbitrary integer $l$ (irreducible representations). (3) Tuples of general direct-sum representations (Sec. 3.4). Therefore, our contribution is a unified framework capable of handling (mixed) representations for *any* given input/output types, far exceeding the scope of rewriting vector-based conclusions.
>
> To the best of our knowledge, no existing work in the classical literature or recent machine learning research provides a unified construction method that simultaneously handles tuples of Cartesian tensors of arbitrary ranks and tuples of spherical tensors of arbitrary types. This capability is particularly crucial for complex scientific tasks beyond simple vector inputs/outputs, such as predicting electronic structure Hamiltonians or density matrices, which require the handling of higher-order spherical tensors.
>
> > **W2.** Related Work Placement and Comparison with Pearce-Crump' work
>
> We thank the reviewer for pointing out the placement issue and for bringing the work of Pearce-Crump [1] to our attention. We accept the suggestion. We have moved the Related Work section from the Appendix to the main text (Sec. 6) to better contextualize our contributions.
>
> We respectfully clarify that work [1] does not surpasses our work due to fundamental differences in scope:
>
> 1. Work [1] focuses on constructing a basis for **linear** equivariant maps. In contrast, our work focuses on constructing generators for **polynomial** invariants and equivariants. Furthermore, our framework can approximate **any continuous non-linear invariant function** (via the Weierstrass approximation theorem ).  Our framework offers a broader generalization where linear maps are just a **special case**, and our approach is equally **constructive**.
> 2. Work [1] primarily treats Cartesian tensors. However, our framework provides a unified construction for **Spherical Tensors (irreducible representations)** and Cartesian tensors(Sec. 3.2). In many scientific domains (e.g., quantum chemistry, electronic structure), data naturally appears as spherical tensors (e.g., $l=0, 1, 2$ orbitals).
>
> 3. Regarding the restriction to $SO(3)$, we focus on this group because it is the primary symmetry group for 3D physical geometry.  For SO(3), the irreducible decomposition and Clebsch–Gordan coefficients are fully known, which allows us to work comfortably with higher-degree irreps (spherical tensors). However, for general $SO(n)$ or other groups, Clebsch-Gordan coefficients are typically unknown.
>
> We have added work [1] in the revised Related Work section (Sec. 6).

---

> > ### Author Response · Authors · 2025-11-24
> > **Response to Reviewer 5UcL Part 2/2**
> >
> > > **W3.** Section 4.1 needs to be much clearer
> >
> > We thank the reviewer for pointing out Section 4.1 was not sufficiently clear,  We have revised Section 4.1 to address these issues. We have added an introductory paragraph to Section 4.1 explicitly interpreting the meaning of the diagrams:
> >
> > "We represent the scalar invariants as fully contracted networks involving the input features $h_i, h_j$ and the message $m_{ij}$. In the diagrams below, nodes represent the input/output quantities, and connecting lines represent the contraction of indices with the identity tensor $\delta_{ij}$ or the Levi-Civita tensor $\epsilon_{ijk}$. Specifically, the red lines denote the output space corresponding to the indices of the message $m_{ij}$ before the final contraction to a scalar. By taking the derivative with respect to $m_{ij}$ (removing the red lines), we obtain the equivariant functions. For simplicity, we just construct the tensor networks that $h$ and $m_{ij}$ appear at most once. More complicated operations can be constructed in the similar way."
> >
> > > **W4.** The strength of this paper
> >
> > We understand the desire to see more diverse applications. We respectfully clarify that the main contribution of this paper is the **theoretical framework** itself, rather than empirical performance on specific benchmarks. Our goal is to solve the problem of systematically constructing valid invariant and equivariant operations for arbitrary data types ((mixed) Cartesian and spherical tensors). This provides a mathematical guarantee of correctness and completeness that ad-hoc designs lack.
> >
> > >**W5.** Improve the paper's readability.
> >
> > We apologize for the oversight. We have conducted a thorough proofreading of the revised manuscript to correct grammatical errors and improve readability.
> >
> > >**Q1.** Comment on the weaknesses
> >
> > Please refer to our detailed responses to Weakness above, where we have addressed these points extensively.
> >
> > >**Q2.** Explain/demonstrate applications
> >
> > We thank the reviewer for this question. Our woks provides a general-purpose methods for constructing equivariant layers for complex scientific tasks, which including: predicting the polarizability tensor (a rank-2 symmetric tensor) or susceptibilities tensor (rank-3 or rank-4 tensor), predicting electronic structure Hamiltonians or density matrices (require handling of higher-order spherical tensor). Furthermore, in generative modeling of materials or molecules, one often wants to condition the generation on specific tensorial properties, such as Born effective charges (a rank-2 tensor ) or stress tensors. Our framework allows to construct the invariant/equivariant interaction terms between the conditioning tensor (e.g. Born charges) and the geometric positions, ensuring the generative process respects the symmetry.

---

> > > ### Comment · Reviewer_5UcL · 2025-11-26
> > >
> > > I have read the rebuttal in full, as well as all of the other reviewers' comments (and the authors' replies to those comments). I agree with the other reviewers that understanding the changes to the manuscript (in full) is difficult without any highlighting of those changes, but I have done my best.
> > >
> > > I welcome the movement of the related work section and accept the comments that the present work is different in scope with [1]. I also appreciate the additional paragraph for Section 4.1 to improve the clarity of the paper.
> > >
> > > However, I maintain that the claims being made regarding the present work's novelty are far too broad, and I see that my view is shared by Reviewer MLAE. I also share the same reviewer's concern regarding "notic[ing] a concerning inconsistency: when asked for more experiments, the authors emphasize the theoretical nature of the work, yet when theoretical clarity is requested, the responses become vague. This is not acceptable." I think Reviewer nQYv's view is spot on: "[t]he authors should demonstrate the value of their proposed architecture via experiments, even if to establish simple conclusions." Without this, the paper's contributions come across as being insubstantial.
> > >
> > > Regarding improving the paper's readability, I can still see on immediate viewing of the updated manuscript many typos, grammatical errors, and poor formatting. I maintain that this detracts from the overall reading experience and needs revision.
> > >
> > > I think the authors need to make (the scope of) their contributions clearer/explicit in the paper itself, and need to provide (simple) experiments to support their theoretical framework. Without this it is difficult to support the publication of this work as it presently stands. Consequently I maintain my current score.

---

### Official Review · Reviewer_MLAE · 2025-10-18

**Soundness:** 2
**Presentation:** 1
**Contribution:** 1
**Rating:** 2
**Confidence:** 5

**Summary:**

The paper proposes a general framework for constructing SO(3)-invariant and equivariant operations using symmetric tensor networks. The authors further illustrate how this applies to geometric GNNs by manually deriving some message-passing operators.

**Strengths:**

- Studying efficient and reliable equivariant/invariant operators is a grand goal.
- The graphical representation constructed in this paper is very interesting and seems to have some relevance to Penrose graphical notation.

**Weaknesses:**

> **Confusing Writing Logic.**

If I understand correctly, the core contribution of this article lies in the Tensor Diagrams (TD) notation it provides (since I haven't seen any novel equivariant/invariant constructions).

Therefore, the appropriate overall layout should be: briefly introduce the definitions of Cartesian and spherical tensors and formalize equivariance, then directly provide a systematic introduction to TD, followed by a discussion in conjunction with geometric graph neural networks.

However, the article currently begins with a lengthy discussion of lemmas, which is extremely unfriendly to those outside the field and extremely trivial and boring to those within it, making it difficult to quickly grasp the article's contributions.

Below, I will specifically address the shortcomings of each paragraph.

> **Overclaim of Contribution.**

In the abstract, the authors claim that their work can "construct valid invariant and equivariant operations", but looking at the entire paper, it seems that this paper is simply reformulating existing work. Of course, this is also a good contribution, but it is necessary to revise the statement of contribution.

In addition, the formalism in this paper makes extensive use of cross products, so it may be very difficult to generalize to O(3) groups. Therefore, I think the authors need to emphasize the scope of application of this paper at the beginning.

> **Lack of Relevant References**

This paper lacks references in several areas. On the one hand, it lacks an introduction to the origins of TD, and on the other hand, it lacks information on geometric graph neural networks. In conjunction with my previous suggestion to the authors to rewrite the paper, I list the discussion that should be supplemented below.

Regarding TD, relevant introductions and discussions such as Penrose graphical notation should be supplemented, and at least the differences between the TD in this paper and Penrose graphical notation should be clearly explained. The following references are recommended:

> - [A] The Tensor Cookbook

Regarding geometric graph neural networks, I suggest a discussion and summary similar to MPNN [B], which should be divided into the following parts:
- From EGNN [C] to HEGNN [D]: Generate scalars through inner products, from 1st-degree irreducible representation to higher-degree irreducible representation.
- From TFN [E] to MACE [F]: Construct and screen through tensor product, from two-body action to multi-body action.
- From LieConv [G] to Frame-Averaging [H]: Construct equivariants/invariants through group convolution, from continuous group convolution to discrete group convolution.
- E2former [I]: Relate Cartesian and spherical tensors and achieve acceleration.
- Theoretical articles [J-N]: used for simple theoretical discussions to illustrate the rationality of the construction.

> - [B] Neural Message Passing for Quantum Chemistry
> - [C] E(n) Equivariant Graph Neural Networks
> - [D] Are High-Degree Representations Really Unnecessary in Equivarinat Graph Neural Networks?
> - [E] Rotation- and translation-equivariant neural networks for 3D point clouds
> - [F] MACE: Higher Order Equivariant Message Passing Neural Networks for Fast and Accurate Force Fields
> - [G] Generalizing Convolutional Neural Networks for Equivariance to Lie Groups on Arbitrary Continuous Data
> - [H] Frame Averaging for Invariant and Equivariant Network Design
> - [I] E2Former: An Efficient and Equivariant Transformer with Linear-Scaling Tensor Products
> - [J] Scalars are universal: Equivariant machine learning, structured like classical physics
> - [K] On the Universality of Rotation Equivariant Point Cloud Networks
> - [L] On the Expressive Power of Geometric Graph Neural Networks
> - [M] Universally Invariant Learning in Equivariant GNNs
> - [N] On the Completeness of Invariant Geometric Deep Learning Models

> **Insufficient experiments**

The experiments in this article are unimportant and seem to have been forced in by the authors to fill the main text. Their inclusion here is a major flaw and should be removed (rather than moved to the appendix). Specifically, because the objective function is complex, we are uncertain which features are needed and which are not. Therefore, when there are very few features, adding features is likely to improve model performance, while when there are many features, adding features will actually reduce performance. This is common knowledge and requires no additional experimental explanation.


---

Overall, as a submission intended to systematically reformulate the field, this paper currently falls far short of the standard necessary for acceptance. The vast majority of its content either simply reiterates domain knowledge or lacks integration with existing work. Both its logical presentation and depth of thought are far below the community average, so I recommend rejection.

Considering that ICLR has accepted resubmitted PDFs in previous years and that authors are not required to perform additional experiments, I believe the above revisions are minimal and can be completed during the discussion phase. If the shortcomings I've raised are addressed, I will reconsider my score.

**Questions:**

See Weakness.

---

> ### Author Response · Authors · 2025-11-24
> **Response to Reviewer MLAE Part 1/2**
>
> We thank the reviewer for the detailed feedback. Below we respond point-by-point.
>
> > **W1.** **Confusing Writing Logic.**
>
> We thank the reviewer for the feedback regarding the manuscript's structure and clarity. We realize there may be a misunderstanding regarding the scope of our contribution, and we would like to clarify our logical flow and core claims.
>
> We respectfully clarify that the core contribution of this work is not merely the Tensor Network diagram notation itself. Rather, our contribution is establishing a systematic theoretical framework capable of constructing any valid $SO(3)$ invariant and equivariant functions for arbitrary inputs/outputs. This includes: (1) Tuples of arbitrary Cartesian tensors (Sec. 3.2), (2) Tuples of arbitrary Spherical irreps (Sec. 3.3), and (3) Tuples of general direct-sum representations (Sec. 3.4). The core contribution lies in the **construction framework** that guarantee the **completeness and correctness** of these operations for such a diverse range of data types, which goes beyond existing works.
>
> We accept your suggestion to improve the accessibility of the introduction. In the revised manuscript, we briefly introduced the definitions of Cartesian and spherical tensors at Sec 2.4. We also note that a detailed introduction to the (symmetry) Tensor Network diagram notation itself is already provided in the Appendix to keep the main text focused on the novel construction rules.
>
> We understand your concern that “begins with a lengthy discussion of lemmas” can feel unfriendly, especially to non-experts. We respectfully argue that this arrangement is natural and essential for our theoretical framework. The lemmas (regarding the generators of vector inputs and symmetric tensors) provide the mathematical foundation on which our main theorems are built. These theorems, together with their explicit tensor network realizations, form the core theoretical contribution of the paper. The discussion on Geometric GNN (Sec. 4) is intended as an **application** of this developed framework, demonstrating how our general theory specializes to concrete Geometric GNN instances. Therefore, placing the theoretical derivations (lemmas and theorems) before the application to GNNs follows a deductive logic essential for establishing the validity of our construction method.
>
> > **W2.**  **Overclaim of Contribution.**
>
> We thank the reviewer for these comments. We realize that we need to be more precise about our specific contributions and the applicable scope. We respectfully clarify that our work is **not** a reformulation of existing results. Instead, our core contribution is providing a systematic, constructive framework that can generate any $SO(3)$ invariant and equivariant map for a wide class of input–output types. To the best of our knowledge, no existing work provides a unified construction method that simultaneously handles tuples of Cartesian tensors of arbitrary ranks and tuples of spherical tensors of arbitrary types. This capability is particularly crucial for complex scientific tasks beyond vectors input–output, such as predicting electronic structure Hamiltonians or density matrices, which require handling of higher-order spherical tensor.
>
> We had explicitly emphasized the $SO(3)$ scope at the Introduction. While the current formalism targets $SO(3)$, it can indeed be generalized to the full $O(3)$ group (which includes inversion). The use of cross products (Levi-Civita tensor) in tensor networks corresponds to the presence of pseudo-tensors. This does not in itself preclude extension to $O(3)$, but it does require tracking parity.  We can assign to every input, output, and structural tensor a parity label (odd or even), so that each tensor network has a global parity given by the product of all its constituent parities. $O(3)$-invariant polynomials are then obtained by restricting to tensor networks with total parity even. We can follow the Theorem 3.6 and remove the output tensor from such a invariant network yields an  $O(3)$-equivariant polynomials that correctly respects the parity of the target output quantity.  In the revised manuscript, we added a discussion in the Conclusion section outlining how the framework extends to $O(3)$ via parity labeling.

---

> > ### Author Response · Authors · 2025-11-24
> > **Response to Reviewer MLAE Part 2/2**
> >
> > > **W3.** **Lack of Relevant References**
> >
> > We thank the reviewer for pointing out the missing context on tensor diagrams and geometric graph neural networks. We will revise the manuscript accordingly. We had explicitly acknowledge the origins of Tensor Network graphical notation in the Preliminaries Sec 2.1 in the revised manuscript. The roots of this diagrammatic notation (in physics) can be traced back to the work of Roger Penrose in the 1970s [1]. Tensor networks share essentially the same fundamental diagrammatic language with Penrose’s graphical. The Symmetric Tensor Networks employed in this work can be understood as a specialized instance of Tensor network applied to group representation theory.  In the revised manuscript,  we have also relocated the Related Work section  from the Appendix to the main body of the paper to ensure a thorough discussion of the background and related literature. We also rewritten the related work about the  geometric GNN (Sec.6) to incorporate structured discussion and added some references you recommended.
> >
> > - [1] Roger Penrose. Applications of negative dimensional tensors. Combinatorial Mathematics and its Applications, Academic Press, 1971.
> >
> > > **W4**. **Insufficient experiments**
> >
> > We thank the reviewer for the detailed comments on the experimental section. We have reflected on this and fully accepted your suggestion. We would like to briefly clarify that the original purpose of the experiments was twofold: (1) To serve as a tutorial demonstration, illustrating how to apply our theoretical framework to a concrete physical problem. (2) To demonstrate the data efficiency of the constructed equivariant models compared to non-equivariant baselines (MLP)  rather than simply exploring feature selection. However, we agree that these experiments are not essential for the main theoretical contribution and including them might distract from the paper's core message, which is the theoretical construction framework. Therefore, we have removed the experimental section from the manuscript to keep the focus strictly on the theoretical framework and the systematic construction of tensor network generators.

---

> > > ### Comment · Reviewer_MLAE · 2025-11-25
> > >
> > > I will first provide a brief response to confirm that I have received the rebuttal. I strongly recommend that the authors highlight all revised parts in the manuscript; otherwise it is extremely difficult for reviewers to quickly locate the changes.
> > >
> > > In addition, I must point out the following issues:
> > >
> > > In the response to W2, the claim “To the best of our knowledge, no existing work … generate any SO(3) invariant and equivariant map for a wide class of input–output types” is incorrect. The statement is overly broad, and in many cases such maps are trivially determined.
> > > - For the former part: for instance, given a second-order tensor (e.g., a covariance matrix), there is no SO(3) map that sends it to a first-order vector (because the principal axes are not SO(3)-equivariant; PCA is not SO(3)). The only possible output is the zero vector.
> > > - For the latter part: given 3D coordinates, constructing arbitrary tensor orders is straightforward, including transforming between Cartesian and spherical tensors. This is standard mathematical knowledge understood for centuries; I do not see how this constitutes a contribution. The authors may refer to SphericalTensor and CartesianTensor on e3nn.io.
> > >
> > > The response to W3 is extremely superficial.
> > > My concern is that the theoretical contribution of the paper is fairly trivial—arguably even below the level of a tutorial—because with only input/output representations, even beginners can perform such analysis. I explicitly requested a substantial rewrite addressing how the framework handles different tensor-product paths and how it deals with complex multi-body interactions. These should be the core of the paper, yet they are not discussed at all. Without these elements, the value of the proposed theoretical framework is unclear.
> > >
> > > Moreover, after reading the responses to other reviewers, I noticed a concerning inconsistency: when asked for more experiments, the authors emphasize the theoretical nature of the work, yet when theoretical clarity is requested, the responses become vague. This is not acceptable.
> > >
> > > Let me give a simple example. In the response to Reviewer 5UcL (W2), the authors claim that the framework can “approximate any continuous non-linear invariant function.” In reality, this deeply depends on message-passing choices and graph construction; it is not something that can be asserted casually. The authors introduce a set of “generators” but do not explain how these generators are actually obtained. This simply shifts the difficulty elsewhere without solving it.
> > >
> > > Overall, I am not satisfied with the current rebuttal and maintain my original evaluation. Unless the authors properly address the issues raised above, I will not support revision.

---

> > > > ### Comment · Reviewer_MLAE · 2025-11-27
> > > >
> > > > I would like to clarify my comment on the experimental section, as it appears to have been misunderstood. My statement that “the experiments are unimportant” referred specifically to the fact that, in the original submission, the experimental design and conclusions were not sufficiently meaningful to validate the claims of the paper. For this reason, I suggested their removal in that context.
> > > >
> > > > I have consistently held the view that experimental validation is necessary for this work, and in this respect I fully agree with the concerns raised by Reviewers 5UcL and nQYv. My original remark should not be interpreted as implying that experiments are unnecessary; rather, the issue lies in the execution and relevance of the original experiments, not in the need for experimental evidence itself.

---

> ### Author Response · Authors · 2025-11-27
> **Response to Comments**
>
> We thank the reviewer for the feedback.
>
> > On Manuscript Highlights
>
> We have uploaded a new revision with changes highlighted in blue to facilitate your review.
>
> > Clarification on  our Claims
>
> We respectfully point out two critical misunderstandings in the reviewer’s assessment regarding the "triviality" of these maps.
>
> (1) On the Existence of Rank-2 to Vector Maps
>
> The reviewer claims that "there is no SO(3) map that sends a second-order tensor to a first-order vector," citing the covariance matrix as an example. This statement holds **only** for *symmetric* tensors (like the covariance matrix $\Sigma$). Indeed, the contraction with the Levi-Civita tensor vanishes for symmetric inputs ($\epsilon_{ijk}\Sigma_{jk} \equiv 0$). However, for a **general** rank-2 tensor $T$ ， the map $v_i = \epsilon_{ijk}T_{jk}$ constitutes a valid, non-trivial $SO(3)$-equivariant map a second-order tensor to a (axial) vector (also see Sec 4.1). Our framework correctly generates this general map.
>
> (2) On "Straightforward Construction from 3D Coordinates"
>
> The reviewer suggests that since tensors can be constructed from 3D coordinates (or decomposed into vectors), our contribution is merely "standard mathematical knowledge." We suspect the reviewer implies that any tensor input can be decomposed into a set of vectors (e.g., via rank-1 decomposition), thereby reducing the problem to vector invariants. However，such decompositions (e.g., finding eigenvectors) are **not** polynomial operations. They involve solving roots of characteristic equations, leading to non-linear and often multi-valued functions .  Our work operates within the framework of Classical Invariant Theory, which seeks the **generators** of the polynomial ring of invariants directly from the tensor components, *without* relying on non-linear or multi-valued decompositions.  We hope these clarifications demonstrate that our work distinct from standard tensor definitions or simple coordinate transformations.

---

> ### Comment · Reviewer_MLAE · 2025-11-27
>
> I must emphasize that I am referring to a specific counterexample: for an inversion-invariant second-order tensor, it is impossible to obtain a corresponding non-trivial first-order inversion-equivariant output. This directly contradicts the authors’ claim that one can "generate **any** SO(3) invariant and equivariant map".
>
> Furthermore, I have serious concerns about the authors’ basic familiarity with the literature in this area (a brief search on Google Scholar already reveals a substantial body of related work), the logical coherence of the rebuttal, and the use of elaborate but largely uninformative physics notation. In my view, the theoretical analysis in the paper falls short of the standard of rigor expected in this field, and several of the main conclusions appear to rest on informal narrative arguments rather than on clearly stated definitions and rigorous proofs.
>
> In addition, I find the authors’ repeated attempts to sidestep or rhetorically reframe these central issues, rather than addressing them directly, deeply unsatisfactory. This repeated evasiveness has exhausted my willingness to engage in further back-and-forth discussion. Until the concerns I have raised are substantively resolved — through precise definitions, rigorous proofs, and appropriately designed experiments — I do not intend to provide further responses. Given the absence of any new experimental validation and the unresolved foundational problems outlined above, I have accordingly downgraded my Soundness score to 1.

---

### Official Review · Reviewer_nQYv · 2025-10-23

**Soundness:** 2
**Presentation:** 2
**Contribution:** 2
**Rating:** 4
**Confidence:** 4

**Summary:**

This paper presents a paradigm for representing equivariant neural networks using tensor network notation. For the specific case of the group $G = SO(3)$, the author discusses the structure of the equivariant neural network when expressed as a tensor network. This provides a methodology for designing networks intended for scientific problems that exhibit corresponding symmetries.

**Strengths:**

The author introduces notation from tensor networks in quantum many-body theory, representing symmetric polynomials as contractions of symmetric tensors. The modeling of equivariant operators is achieved by representing these symmetric tensors with tensor networks. This demonstrates the author's solid theoretical foundation, and the content is presented with relative clarity.

**Weaknesses:**

1. Lack of graph-related experiments. The author mentions equivariant graph neural networks in the title of Section 4, and the proposed method is also aimed at graph neural networks. However, in the experiments in Section 5, the author only conducted experiments related to the prediction of constitutive relations in solid mechanics, with no experiments related to graphs. The author also specifically emphasizes applications on geometric graphs in the contributions, so I believe it is necessary for the author to add relevant content.

2. Contribution is somewhat lacking, and the discussion is not thorough. The author's discussion of $G = SO(3)$ is built upon the structure of $SO(3)$-invariant polynomials, a topic that can be found in [1]. The author rewrites the conclusions regarding invariant polynomials into the tensor network structure of Theorem 3.3. However, after this reformulation, the author does not delve into how this new representation connects different equivariant neural networks or what effective ideas it can bring to network design, which is a topic of interest to many readers. The author should mention the graphical representations of existing equivariant neural networks under this paradigm. Furthermore, based on this paradigm, the author could proceed to discuss the design of new network structures and compare results before and after the modifications. This would make the paper much more substantial.

3. Some imprecise statements and missing explanations. For example, in Section 3.2, lines 179-180, higher-order tensors are mentioned. The author states that for higher-order tensors, these generators would not be minimal. However, these generators include the cross-product symbol, and the definition of a cross product between higher-order tensors is not straightforward. The author should revise the relevant statement. For instance, the notation $(1)^{\otimes l}$ is ambiguous; it should represent a type of tensor or tensor space. But in lines 209-212, where $P_{l}(\boldsymbol{x})$ is a tensor network, the statement that $P_{l}(\boldsymbol{x})$ becomes the tensor $(1)^{\otimes l}$ appears imprecise. Perhaps the author means that the output of $P_{l}(\boldsymbol{x})$ is of type $(1)^{\otimes l}$. As another example, in the tensor network diagrams in Section 3.3, the meaning of the numbers on the edges is unclear. They might indicate that an index from each of the two tensors corresponds to the same representation space, thus allowing contraction. This requires clarification from the author.

4. Suggestion for notational change. In Definition 2.2, the summation indices for the direct sum spaces of both the input and output are $i$. One of them could be changed to $j$.

Overall Assessment: In my personal view, this is an interesting paper that could be extended in meaningful ways. For instance, after representing equivariant neural networks as graphs, one could emulate [2] to perform a search for network architectures to find better-performing models. However, the author's exploration of the problem stops at merely reformulating the network structure. This limits the discussion to a superficial level of representation. As it stands, I believe the paper may not meet the standard for acceptance. If the author were to conduct a more in-depth discussion building on this foundation, I believe this could become a paper with high potential and a significant impact on the community.

Reference:

[1] Villar, Soledad, et al. "Scalars are universal: Equivariant machine learning, structured like classical physics."

[2] Elsken, Thomas, et al. "Neural architecture search: A survey."

**Questions:**

See weaknesses.

---

> ### Author Response · Authors · 2025-11-24
> **Response to Reviewer nQYv Part 1/2**
>
> We sincerely thank the reviewer for the insightful and constructive feedback. Below, we address the specific concerns and detail the revisions made to the manuscript.
>
> >**W1.** Lack of graph-related experiments.
>
> We thank the reviewer for raising this point. As stated in the paper, our primary goal is to establish a systematic theoretical framework capable of constructing *any* valid $SO(3)$ invariant and equivariant operation for arbitrary inputs/outputs. Our framework can be applied to construct valid equivariant message-passing operator of equivariant graph neural networks, given the forms of node features and messages. The contributions of our method is that it generates the **candidate space** of valid message-passing operators. However, we do not know a priori which specific operators or combinations of them will lead to performance improvements for a specific graph task. Manually selecting specific operators to compete on specific graph task for a given dataset is an empirical question of architecture design/search, rather than a theoretical question of operator construction.
>
> We agree with your insightful comment in the "Overall Assessment" that combining our method with Neural Architecture Search (NAS) is a promising direction. Our framework essentially defines the **search space** for such an algorithm. We have revised the Conclusion section to explicitly discuss this, highlighting that our work provides the theoretical engine to enable the future NAS work you suggested.
>
> >**W2.** Contribution is somewhat lacking, and the discussion is not thorough.
>
> We thank the reviewer for this opportunity to clarify our contributions and expand the discussion. We respectfully clarify that our work is **not** merely a reformulation of the conclusions in [1] by tensor network. While Ref. [1] primarily treats inputs and outputs that are **tuples of vectors** (rank-1 Cartesian tensors), our framework is more general. Our method systematically handles (1) **Tuples of arbitrary Cartesian tensors:** As detailed in Sec. 3.2, we handle inputs of any rank (e.g., matrices, rank-3 tensors), not just vectors. (2) **Tuples of arbitrary spherical irreps:** In Sec. 3.3, we provide explicit tensor network generators for inputs labeled by arbitrary integer $l$ (irreducible representations). This covers the foundation of methods like TFN. (3) **Tuples of general direct-sum representations:** In Sec. 3.4, we further generalize this to general representations. Therefore, our contribution is a unified framework capable of handling (mixed) representations for any given input/output types, far exceeding the scope of rewriting vector-based conclusions.
>
> We fully agree that connecting our formalism to existing architectures is valuable. In fact, the primary objective of Sec. 4 is precisely to leverage our formalism to provide graphical representations for existing equivariant neural networks. These include EGNN-style vector operations (sum, dot product, cross product)[1-4], Cartesian tensor message passing[5], and the Tensor Product (TP) operation in TFN-style[6-7]. To make this connection explicit, we have added the relevant references in Sec. 4 in the revised manuscript.
>
> - [1] Víctor Garcia Satorras, et al. E(n) equivariant graph neural networks, 2021
> - [2]  Kristof Schütt, et al. Equivariant message passing for the prediction of
>   tensorial properties and molecular spectra, 2021.
> - [3]  Congyue Deng, et al.
>   Vector neurons: A general framework for so(3)-equivariant networks, 2021
> - [4]  Tuan Le, et al. Equivariant graph attention networks for molecular property
>   prediction, 2022
> - [5] Junjie Wang, et al. E(n)-equivariant cartesian tensor message passing interatomic potential, 2024
> - [6] Nathaniel Thomas, et al. Tensor field networks: Rotation- and translation-equivariant neural networks for 3d point clouds, 2018
> - [7] Maurice Weiler, et al. 3d steerable cnns: Learning rotationally equivariant features in volumetric data.

---

> > ### Author Response · Authors · 2025-11-24
> > **Response to Reviewer nQYv Part 2/2**
> >
> > >**W3.** Some imprecise statements and missing explanations.
> >
> > We appreciate the reviewer's close reading and pointing out of unclear statements. We clarify those points as follows:
> >
> > For vector inputs, the minimal generating set is  $\{x_i\cdot x_j, (x_i\times x_j)\cdot x_k\}$ .  For higher-rank Cartesian tensors, we no longer talk about cross products of those tensors, which is not well-defined; instead, Theorem 3.3 shows that any invariant polynomial is generated by connecting input tensors with $\delta_{ij}$ and at most one $\epsilon_{ijk}$.
> >
> > The $(1)^{\otimes l}$  denotes the representation space $V^{\otimes l}$. Our intended meaning is that $P_l(x)$ is an element of the representation space $(1)^{\otimes l}$. In the revised manuscript, we have clarified in Sec. 3.3 that $(1)^{\otimes l}$ denotes the $l$-fold tensor-product representation space and replace "Then $P_l(x)$ becomes a tensor $(1)^{\otimes l}$" with "Then $P_l(x)$ becomes a tensor in the space $(1)^{\otimes l}$."
> >
> > In Sec.3.3, the numbers on the edges represent the irreducible representation type of the vector space associated with that index. In the revised manuscript, we added an explicit explanation in Sec. 3.3.
> >
> > >**W4.** Suggestion for notational change. In Definition 2.2, the summation indices for the direct sum spaces of both the input and output are $i$. One of them could be changed to $j$.
> >
> > We have adopted this suggestion. In the revised manuscript, we changed Definition 2.2 to use $\bigoplus_j V_j \to \bigoplus_i U_i$ and adjust the subsequent notation accordingly.

---

> ### Comment · Reviewer_nQYv · 2025-11-25
>
> I appreciate the authors' detailed response. Regarding the rebuttal, my comments are as follows:
>
> - **Regarding graph-related experiments:** I maintain my position that the authors must include graph-related experiments, even if it is just a simple ablation study on the graph architecture. Given that the paper devotes considerable space to discussing equivariant message passing operators which is a topic of significant interest to the machine learning communit, I consider these experiments crucial. I acknowledge the authors' point that priors for specific task-based graph neural network designs are often unknown, however, it is the authors' responsibility to derive insights through experimentation. The authors should demonstrate the value of their proposed architecture via experiments, even if to establish simple conclusions.
>
> - **Regarding network architecture search (NAS):** I previously suggested that the authors could consider using search-based methods for architecture design, as this is a valid approach to network design. In the revised version, the authors merely discussed this in the Conclusion section. Without experimental support, this addition is insufficient to support this contribution.
>
> - **Regarding the analysis of existing models:** To better demonstrate how existing equivariant neural networks are incorporated into the authors' framework, it would be beneficial to represent these existing architectures using tensor networks. This would facilitate a clearer elaboration of the guidelines for designing equivariant neural networks.
>
> - **Regarding formatting:** In the revised version, changes made to the manuscript should be clearly marked. Otherwise, it is difficult for reviewers to locate and evaluate the work done during the rebuttal period.
>
> In summary, to better demonstrate the theoretical value of the work and to provide better guidance for the machine learning community, I believe a clear methodology for guiding equivariant graph neural network design needs to be proposed and emphasized in the paper. The theoretical nature of the paper is not a valid justification for omitting relevant experiments. Therefore,  I maintain my current rating.

---

### Meta-Review · Area_Chair_PT2V · 2025-12-22

**Summary:**

The paper introduces a method to construct invariant/equivariant maps to/from tuples of cartesian/spherical tensors, together with a graphical representation similar Penrose's.

Reviewers found the formulation interesting and potentially useful, but serious concerns with raised about the significance of the contributions, relation to prior work, missing references, writing/organization, and lack of experiments/practical applications.

Reviewers were unanimously negative and the rebuttal was insufficient. Thus, I recommend rejection.

**Reviewer Concerns:**

The most serious concerns were about lack of significance and novelty, and relation with prior works -- this was raised by all reviewers. The rebuttal attempts to clarify the contributions but they do seem limited and not significant enough, specially considering the missing discussions about related work.

Reviewers were unhappy with the significance of the theoretical contributions, but mentioned that more experiments and practical applications could strengthen the submission. Unfortunately, this was not addressed by the rebuttal.

**Reviewer Scores:**

The rebuttal was not convincing so I don't believe any reviewer would change their rating.

---

### Decision · Program_Chairs · 2026-01-26

Reject